# Perceived benefits and challenges one year after receiving brief therapy in a district psychiatric centre. An exploration of patients' and GPs' experiences: A qualitative study

Hilde V. Markussen[1,2]*, Lene Aasdahl[3,4], Petter Viksveen[5], Marit B. Rise[2,6]

1 Faculty of Medicine and Health Sciences, Department of Mental Health, Norwegian University of Science and Technology, Trondheim, Norway, 2 St. Olavs Hospital, Trondheim University Hospital, Nidaros District Psychiatric Centre, Trondheim, Norway, 3 Faculty of Medicine and Health Sciences, Department of Public Health and Nursing, Norwegian University of Science and Technology, Trondheim, Norway, 4 Unicare Helsefort Rehabilitation Centre, Rissa, Norway, 5 Faculty of Health Sciences, Department of Quality and Health Technology, SHARE–Centre for Resilience in Healthcare, University of Stavanger, Stavanger, Norway, 6 Faculty of Medicine and Health Sciences, Department of Mental Health, RKBU Central Norway, Norwegian University of Science and Technology, Trondheim, Norway

* hilde.v.markussen@ntnu.no

**Data Availability Statement:** There are both ethical and legal restrictions on sharing the data set due to sensitive information. It is not possible to publish

## Abstract

### Background

Scarce treatment resources put pressure on mental health services prompting innovations in service provision. Various innovative strategies have been introduced to provide patients with improved and effective treatment due to increased demands for mental health services. In 2016 a district psychiatric centre (DPC) started a brief treatment program to provide early and effective help for moderate depression and anxiety. There is little evidence regarding the potential benefits that different mental health patients may experience after receiving brief therapy treatment. Moreover, few studies have explored the experiences of referring general practitioners (GPs) with different patient outcomes after brief therapy. The aim of this study was to investigate the long-term experiences of patients who received brief therapy at a DPC at least one year ago, as well as the experiences of general practitioners (GPs) who referred patients for such treatment since 2016. Specifically, the study sought to determine patterns in the patients' stories and GPs' experiences to see if it could provide new insight for further studies.

### Methods

We conducted individual interviews with a total of seventeen participants, consisting of eleven patients and six GPs. Using an exploratory approach, we analyzed patients' narratives as they described them in the interviews, employing inductive and thematic analysis techniques. The GPs' experiences of referring several patients to brief therapy were also subjected to thematic analysis. In addition, we synthesized the patients' experiences into condensed stories for comparison. The experiences of GPs, who had referred patients to this brief treatment program over several years, were compared with the patients'

the original data as participants were guaranteed that their interviews would not be made publicly available. Therefore, data publication would violate their privacy rights and conflict with the General Data Protection Regulation (GDPR). The approved information letter to participants in the present study stated; The researchers have a duty of confidentiality, and information from the interviews will not be given to employees at the DPC nor anyone else. This study was approved by the "Regional Committee for Medical and Health Research Ethics" (REK) in central Norway (2018/49). REK approved the information letter before we used it in the study. Email: rek-midt@mh.ntnu.no The committee secretariats can also be contacted by telephone or office address which can be found at the online portal: https://rekportalen.no.

**Funding:** The first author (HVM) and the last author (Rise MB) received the funding through their affiliation with NTNU and St.Olavs Hospital. The Liaison Committee (NO. Samarbeidsorganet) between the Central Norway Regional Health Authority and Norwegian University of Science and Technology (NTNU) funded the work under Grant no. 22314. The funders website: https://helse-midt.no/samarbeidsorganet. The funders had no role in study design, data collection and analysis, decision to publish or preparation of the manuscript.

**Competing interests:** The authors have declared that no competing interests exist.

experiences and reflections one year after receiving brief therapy. This comparison aimed to challenge and deepen the understanding of condensed patient stories.

## Results

The main results are presented as three condensed patient stories: A) Coping with mental problems; B) A path to another treatment; and C) Confusion and lost hope. The GPs' experiences are included in each condensed patient story to challenge and elaborate on relevant aspects.

## Conclusion

Time-limited brief therapy was experienced as beneficial for patients with moderate affective and anxiety disorders but was experienced as unsuitable for those with more severe conditions. This raises important questions about the appropriateness of offering brief therapy to a diverse patient population and the efficient use of healthcare resources. We recommend further research to enhance understanding and develop tailored treatment services for different ailments. Identifying which patients benefit most from specific therapies can improve outcomes and resource allocation. An important improvement measure might be to enhance early communication between patients, General Practitioners (GPs), and District Psychiatric Centres (DPCs) before referrals. Ensuring brief therapy is targeted to those likely to benefit the most will enhance treatment effectiveness.

Additionally, we suggest implementing joint assessment meetings to facilitate comprehensive information exchange and coordination among different care levels. This approach would improve assessments, treatment planning, and follow-up strategies, ultimately leading to better patient care and resource management.

## Introduction

The demand for mental health services has increased worldwide [1]. Depressive disorder and generalized anxiety disorder are among the most common psychiatric conditions [2], and prevention and treatment of these conditions are of great importance to individuals and society. Individuals with anxiety and depression also have a higher risk for other chronic illnesses, leading to higher healthcare expenditures [3]. These are important arguments for starting effective treatment at an early stage.

Scarce treatment resources put pressure on mental health services to make innovations in service provision and various innovative strategies have been introduced to provide patients with improved and effective treatment [4]. This has led to increased interest in short-term, cost-effective therapeutic approaches. [5]. One of these is time-limited therapies [6], where a limited number of therapy sessions are provided individually or in groups. While the terms "brief therapy", "time-limited therapy "and "short-term therapy" are used interchangeably in the literature [7], brief therapy usually involves cognitive-behavioral therapy (CBT) and solution-focused approaches, which focuses on the present ("here and now") and the patient's resources and strengths [7].

Brief therapy approaches may increase access for patients and cost-effectiveness for society [2]. So far, brief psychodynamic psychotherapy has shown promising benefits for adults with

common mental disorders [8], and brief CBT seems effective in treating depression and anxiety [9]. Brief cognitive-behavioral and mindfulness-based interventions are established for depressive and anxiety disorders [2]. Studies indicate that brief CBT is effective for psychosocial outcomes (e.g., psychosocial adjustment to illness) and promising for behavioral outcomes (e.g., physical activity) in medical settings [9]. Brief therapies have been recommended due to their focus on symptom reduction [10], but it has been suggested that treatment deadlines adversely affect the therapeutic process [11]. Weekly treatment for common mental health problems (4–26 sessions) might increase the rate of improvement compared to rarer schedules [3,12]. Although brief treatment courses are associated with rapid improvement in mental health outcomes [13], an adaptation of treatment length to meet individual needs has been claimed necessary [14], suggesting that a fixed duration of treatment is inappropriate [15]. Furthermore, improvement and recovery are associated with the severity of symptoms and disability at the start of treatment, and some research suggests that a low frequency of initial treatment sessions can lead to an even less favorable outcome and a more chronic course of mental illness [3].

Worldwide, a significant number of individuals with mental disorders do not receive any treatment, and many of those who do, do not benefit from it [16]. So far, little is known about the potential benefits patients may experience following brief therapy treatment in mental health services. In addition, there is little knowledge about referring general practitioners' (GPs) experiences with patient outcomes after receiving brief therapy. Exploring patients' and referring GPs' experiences can provide knowledge about who might benefit from the treatment. Therefore, the aim of this study was to investigate the long-term experiences of patients who received brief therapy at a district psychiatric centre at least one year ago, as well as the experiences of general practitioners (GPs) who referred patients for such treatment since 2016. Specifically, the study sought to determine patterns in the patients' stories and GPs' experiences to see if it could provide new insight for further studies.

## Methods

This qualitative study was inspired by phenomenology [17] and based on individual semi-structured interviews [18] with 11 patients and 6 GPs. We chose an explorative approach and analyzed the patients' narratives from the last year using inductive and thematic analysis techniques. The experiences of the GPs from the interviews became another subject of thematic analysis. The patients experiences were synthesized in condensed stories to clarify the distinctions between their experiences. The experiences of GPs, who had referred patients to this brief treatment program over several years, were compared with the condensed patient stories one year after receiving brief therapy. The comparison aimed to challenge and deepen the condensed patient stories–and to enable hypothesis-generation.

### Study setting

In Norway, health care is organized at two levels, in the primary and secondary health services. Patients can contact their GP for help with mental health problems, and the GP may refer them for specialized treatment at district psychiatric centres (DPCs) at the secondary service level. This study took place at a DPC in Central Norway and was part of a larger research project, exploring patients' and therapists' perceptions of and experiences with brief therapy [19,20].

In 2016, the DPC started a brief therapy program to provide treatment to a growing young patient population with anxiety and depression as the main reason for referral. The time-limited brief treatment program was started after a short trial-period of brief therapy provided in

the general outpatient services at the DPC. The DPC aimed to design a service with a short waiting time to provide effective help at an early stage. At this DPC, brief therapy was described as time-limited cognitive psychotherapy restricted to ten treatment sessions [7]. According to the head of the brief therapy unit, brief therapy included CBT and metacognitive therapy [21] and was offered to adult patients over 18 years of age in general, with moderate to severe anxiety, depression, and other mental disorders. The head of the unit also described the time-limited treatment as more flexible than 10 treatment hours. Depending on the individual patient's needs, it could be shorter or extended by a few hours.

The DPC leadership described that brief therapy was established to provide metacognitive and cognitive behavioral therapy to patients with anxiety and depression as the leading cause of referral. In 2019, a total of 459 patients received brief therapy at the unit. According to the DPC, Brief therapy was not intended to be offered to patients who were considered to need more comprehensive treatment, and if such patients were referred, they could be considered for another more comprehensive or long-term offer at the DPC.

This DPC was part of a university hospital in Central Norway, producing 260 man-years (full-time equivalents). It was one of three similar DPCs within the hospital trust. The catchment area included approximately 110,000 persons in urban and semirural areas and parts of a large city. The DPC provided inpatient treatment, ambulatory services, and different types of outpatient treatment. Patients were mainly referred to the DPC from their GP, or by psychologists or psychiatrists in private practice. The DPC assessed the referrals and decided whether treatment was appropriate or not and offered a right to treatment within a specific time period (a waiting period guarantee).

## Participants and recruitment

A total of 17 informants, 11 patients, and 6 GPs participated in the study. Eligible participants were patients who had received brief therapy in the DPC at least a year ago and GPs who had referred patients to this brief therapy unit. The patients in this study had earlier participated in a qualitative study during treatment [20] and were then informed by the first author that they might be invited for another interview in a second study. The original inclusion criterion was patients who received brief therapy treatment in the period from 2019 to 2020. The authors were not involved in the selection of patients beyond writing the information letter. Patients were informed by the therapists at the DPC about the study while receiving treatment. The therapists provided the first author's contact information for those who consented to take part. The first author then contacted these patients and made appointments for interviews. The patients received verbal and written information about the study and signed a consent form before participating in the first interview. Patients were recruited through purposeful criterion sampling [22], i.e. those who received brief therapy in the defined time-period met the inclusion criteria. In the first study, sixteen patients consented to participate, but patient interviews were terminated when the authors considered that the study had sufficient information power (12 patients). The patients confirmed their consent to participate prior to the second interview. Twelve participants from the first study initially agreed to participate in the second study. However, one participant later informed that she was unable to participate in the second interview due to severe mental illness. Therefore, the final sample included 11 patients in the present study.

GPs were also recruited through purposeful criterion sampling [22]. The inclusion criterion was that they worked at a GP office that referred patients for brief therapy and that they had experiences following the referral of patients for such treatment after 2016. Potential participants were informed about the study by the first author (HVM) via e-mail to the receptionists

at several GP offices in the geographical area close to this DPC. Reminders were sent. The participants responded to the e-mail confirming that they would participate, and the first author called them to make an appointment for an interview. The GPs received verbal and written information about the study and signed a consent form before participating in the interview.

## Data collection

Data from patients and GPs were collected in the period from early January to late March 2021.

We chose an exploratory approach with a broad research question, giving patients the opportunity to tell their personal stories of the period following brief therapy. We were careful not to set too strict boundaries for interview topics and rather let patients talk about issues they considered important to share. Two different semi-structured interview guides [18] were used, adapted to patients and GPs, respectively. This included predefined topics, and interviewers probed further as the participants responded. The authors processed the interview guides thoroughly so that the questions were as open as possible, to allow room for the participants' own stories, perspectives, experiences and understanding.

An important topic in the interview guide for patients was a particular focus on participants' history after treatment and experiences of outcomes one year after the end of treatment. Participants were asked whether they considered having perceived any changes due to brief therapy in the past year and how they described their life situation after receiving brief therapy. This encompassed the patients' experiences of personal outcomes from the treatment and their experiences of whether the treatment suited them or not. They were also asked to share their experiences of state of health and whether they had sought or received other mental health services after completing brief therapy.

Key topics addressed in the interview guide for GPs were their experiences with referring patients to brief therapy and any changes they observed in the patients following treatment. In addition, the GPs were asked how they regarded brief therapy as part of the health services for this patient group. Most of the topics in the interview guides were unchanged, but one topic identified in the initial interviews with patients was incorporated into interviews with GPs, focusing on their perspectives of whether the treatment suited the patients who received brief therapy.

The first author (HVM) conducted all the individual interviews. No one else was present during the interview but the participants and the researcher. Due to the covid 19 pandemic, the interviews in the present study were conducted using either online (Skype) technology or telephone, according to the patients' and GPs' wishes. One interview was carried out face to face as this was the only option for the patient. The interviews with patients lasted from 45 to 60 minutes, with one exception (22 minutes). The interviews with GPs lasted 30–40 minutes. The interviewer (HVM) made notes during and after each interview, and all interviews were audio-recorded and transcribed verbatim. The transcripts were not returned to participants for comments and/or corrections, and they could not provide feedback on the findings.

## Data analysis

The data analysis was inspired by phenomenology [17] to describe the informants' experiences as expressed by themselves. The analysis was conducted by four researchers with different backgrounds (medicine, psychology, social sciences, and health services research). HVM was a female doctoral candidate in medicine (now completed) with a professional background in medical sociology and community planning at the master's level (MA) and several years of experience in mental health hospital planning, including service innovation tasks from this

area of Norway. LA is a female associate professor in public health and physician (specialist in physical medicine and rehabilitation). PV is a male researcher and associate professor in health sciences and health services researcher. MBR is a female researcher and professor in mental health work with psychology and public health background.

One researcher (HVM) performed initial empirical coding [23] of all interviews by systematically examining the entire interview material. The intention was to code the material line by line as close to the empirical data as possible without analyzing the content, thereby looking for common patterns present across the patient narratives and GP interviews that could shed light on the aim of the study. The purpose was to establish a coded material that could contribute to the analysis process. In addition, HVM returned to the original empirical material later to select exact quotes from individual interviews.

Inductive analysis of patient narratives [24] was then performed to explore categories and themes across the patients narratives and our conceptualisation included an inductive discovery of categories and later also themes that emerged from the data. Initially, the first author (HVM) read all patient interviews thoroughly by listening through the audio recordings twice while reading the transcripts. All semi-structured patient interviews were then read by the last author (MBR), and six of the patient interviews were additionally assessed by the other two authors (LA and PV three interviews each). The first impressions from patient interviews were discussed between all four authors, resulting in preliminary themes from the patient's experiences. Examples of preliminary themes at this first step were "varying experiences of brief therapy one year after the end of treatment–both positive and negative", "experiences of who brief therapy was suitable for", and "was brief therapy offered in response to patients' individual needs?" As part of the semi-structured interviews, we gathered patient stories, which served as data in the analysis. The analysis resulted in themes across the patients' individual stories. We intended to look beyond the collection and construction of stories, which were the focus of the storytellers/informants. We found that an important advantage of such a narrative analysis (analysis of stories of the patients' last year since brief therapy ended) was to bring order to the experiences by seeing individual experiences in a story as belonging to a category or theme. The first impressions resulted in preliminary themes reflecting the patient stories, and all the four authors then searched for alternative interpretations in several discussions. Secondly, all GP interviews were read by HVM and MBR, and these interviews were also assessed by LA (three interviews) and PV (three interviews). MBR, LA, and PV shared their first impressions and suggested preliminary themes. The suggestions were then collected and shared with all the authors by HVM. Examples of preliminary themes at this second step were "different experiences with referral", "how brief therapy is included in the health service", and "benefits for which patients".

Third, descriptions of preliminary themes and categories across the patients' experiences were contrasted with the preliminary themes from the GPs' interviews. The comparison helped to reinforce the two approaches, such as inductive analysis of patient narratives [24] and comparative thematic analyses of the two groups of participants' experiences. The combined approach could identify patterns across the dataset (themes from patients' stories and GPs' experiences) and help answer the research question [25]. Meaning units were identified, condensed, and coded based on conformity and differences. All the authors contributed to the comparison and sorting of data. Themes were constructed and validated through discussions in the entire author group and were based on common patterns and categories. Empirical dimensions were formed for each interview, and similarities across the interviews were reflected and presented in the composite and condensed stories [26] so that non-verbatim presentations of the participants' experiences became more generally representative accounts of the participants' experiences.

At last, the results were presented as thematically grouped condensed stories [26], reflecting the patients' most meaningful experiences. The process continued until data reached a point of convergence, resulting in three condensed stories that encompassed the main features of the data material. Several patients are reflected in more than one story. In addition, the GPs' experiences were added to the three stories, including their perspectives on patient outcomes. The tension or conflict between different patient narratives became clarifying for analysis and the GPs' experiences helped us to complete the picture. Finally, the authors discussed and concluded that the condensed stories, combined with the GPs experiences, reflected the main topics in the data material. Quotes from the data material were used to elaborate and illustrate the results. They were translated by PV, checked, and approved by MBR and HVM. Quotes are presented with a participant number and a letter indicating the gender of the participant (F: female, M: male). The quotes from the GPs are presented without gender (to ensure that potential identification of participants e.g. by therapists was prevented), but by number to illustrate that different GPs are quoted (GP1 –general practitioner first interview).

Preliminary findings were presented and discussed in research group meetings focusing on patient education and participation and contributed to authors reassessing themes and considering alternative interpretations. For example, patients' experiences associated with the Covid-19 pandemic were considered, but ultimately not included in the data analysis.

The following table (Fig 1) illustrates the analysis process.

## Ethics

This study was approved by the Regional Committees for Medical and Health Research Ethics in Central Norway (2018/49/REK Midt). All participants in this study received verbal and written information about the study and signed a consent form before taking part in interviews. The project was conducted in line with the Helsinki Declaration (World Health Organization, 2010).

## Results

A total of 17 participants took part in individual interviews, 11 patients and 6 GPs. Patient characteristics are described in Fig 2.

### Patient characteristics

The patients were between 20 and 50 years old. Some had several years of experience with specialized mental health treatment, whereas others had none. All received brief therapy in the period from July 2019 to February 2020. The patients reported receiving from 8 to 15 individual treatment sessions. The reasons for referrals were mainly anxiety and depression, but the patients reported multiple diagnoses described in a table (Fig 2).

Almost all patients said they did not know what brief therapy was before receiving the treatment and expressed appreciation for receiving treatment quickly.

### GP characteristics

GPs included four men and two women, aged 35 to 50 years. The six GPs worked in five different GP offices. All had experience with referring patients to this DPC since 2016 and explained that their patients had received either brief therapy or both brief therapy and standard outpatient treatment afterward. Two of the GP offices (represented by 2 GPs) had regular collaboration meetings with the DPC regarding referrals of patients, while three offices had not (represented by four GPs). One of the GPs did not attend collaboration meetings but had experience with them from a previous workplace.

| Two different semi-structured interview guides | |
|---|---|
| Interview guides designed with open-ended questions to allow room for participants' to express their stories, perspectives, experiences, and understanding. | |
| 1) **Interview guide for patients**:<br>Focus on participants' personal stories and self-reported outcomes one year after treatment | 2) **Interview guide for GPs**:<br>Focus on experiences of referral for brief therapy and changes observed in patients one year after treatment |
| 1a) **Examples of preliminary themes from the patient stories/narratives:**<br><ul><li>Varying experiences of brief therapy one year after the end of treatment – both positive and negative</li><li>Experiences of who brief therapy was suitable for</li><li>Was brief therapy offered in response to patients' individual needs?</li></ul>**Analysis**: inductive analysis of patient narratives | 2a) **Examples of preliminary themes from the GPs experiences since 2016:**<br><ul><li>Different experiences with referral</li><li>How brief therapy is included in the health service</li><li>Benefits for which patients</li></ul><br>**Analysis**: thematic analysis. |
| 2) **Comparative thematic analysis**: Preliminary themes and categories across patients' experiences contrasted with preliminary themes from GP interviews. Comparison of the two groups of participants' experiences. | |
| 3) **Meaning units** identified, condensed, and coded based on conformity and differences. | |
| 3a) Examples of **meaning units for patients**:<br><ul><li>Mastering ailments – or not</li><li>Hope for the future – or not</li><li>Cognitive techniques</li><li>Better or worse after treatment</li><li>Important factors for treatment success</li></ul> | 3b) Examples of **meaning units for GPs**:<br><ul><li>Important factors for treatment success</li><li>Referrals and assessments</li><li>Who became better or worse after treatment?</li><li>Lack of other treatment options for patients with long-term ailments</li></ul> |
| 3c) Examples of **common codes** for experiences that:<br><ul><li>Treatment was useful</li><li>Treatment became an extended investigation phase</li><li>Some patients should probably not have received the offer</li></ul> | |
| 4) **Main results presented as thematically grouped and condensed patient stories**, with GPs' experiences, including their perspectives on patient outcomes:<br><ul><li>Coping with mental problems</li><li>A path to another treatment</li><li>Confusion and lost hope</li></ul> | |
| 5) Quotes from the data material used to elaborate and illustrate the results. | |
| 6) Main results/condensed stories described and discussed + conclusions and recommendations. | |

**Fig 1. Illustration, representing the coding process in this study, including some examples of meaning units.**

## Main results

This study explored the experiences of patients who had undergone brief therapy at a district psychiatric centre minimum one year ago and the experiences of general practitioners (GPs) who had referred patients to such treatment. According to both patients and GPs, some patients had benefited from brief therapy, whereas others had not. The results are presented as three condensed patient stories, described in a (Fig 3) which illustrates condensed patient-stories representing the main results after analysis (retrieved from Markussen Hilde V. Doctoral theses at NTNU, 2023:284).

| Variables and categories | N (%) |
|---|---|
| **Gender** | |
| Female | 4 (36) |
| Male | 7 (64) |
| **Age** | |
| 20 – 25 years | 5 (45) |
| 26 – 35 years | 3 (27) |
| 36 – 50 years | 3 (27) |
| **Previous specialist mental health treatment** | |
| Yes | 8 (73) |
| No | 3 (27) |
| **Referred for further treatment after brief therapy** | |
| Yes | 5 (45) |
| No | 6 (55) |
| **Diagnoses (self-reported)** | |
| Depression | 8 (73) |
| Anxiety | 5 (45) |
| Other[a] | 7 (64) |

a. Includes PTSD, personality disorder, bipolar disorder, neurological disorder, unresolved complex disorder.

**Fig 2. Patient characteristics (patients' self-reported information).**

### *Story A.* **Coping with mental problems**

These patients reported brief therapy as beneficial, and that brief treatment suited them and their state of health. They were satisfied with the short waiting time before starting brief therapy and said that brief therapy was precisely what they needed. These patients experienced a mutual connection with the therapist early in the process, which they described as

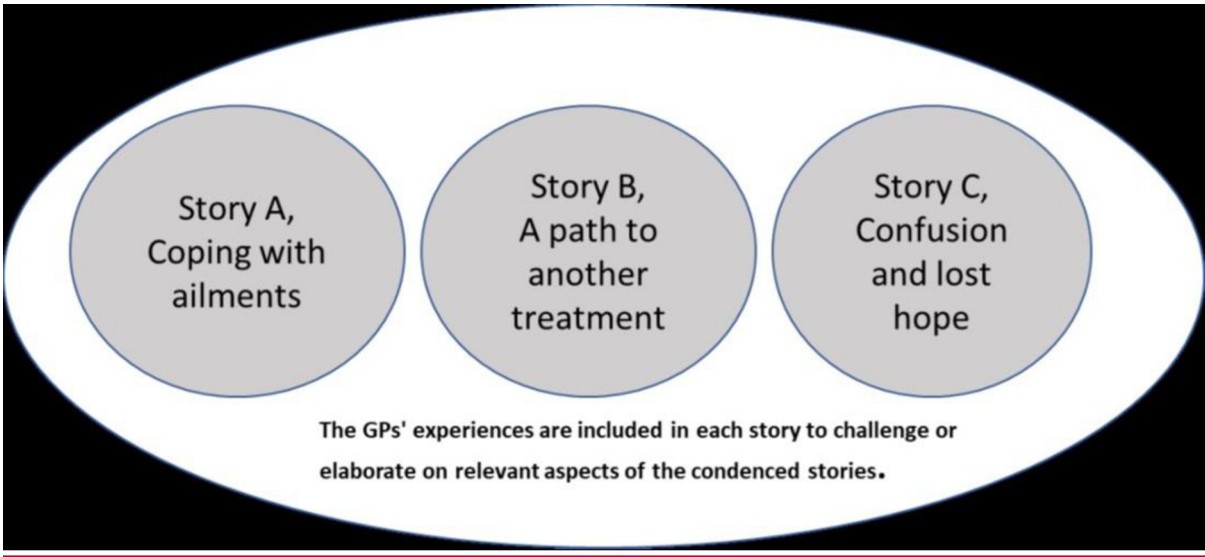

**Fig 3. Condensed patient-stories representing the main results after analysis.** A) Coping with mental problems; B) A path to another treatment; and C) Confusion and lost hope. Some patients are represented in more than one condensed story and the GPs' experiences are included in each story to challenge or elaborate on relevant aspects.

interpersonal acceptance and respect. They said this was crucial as a starting point for cooperation with the therapist on implementing tools from brief therapy. It gave them a feeling of security even though they knew the treatment duration was short. In addition, new knowledge acquired through cognitive exercises and mindfulness training contributed to a greater understanding of their ailments. This new and acquired insight felt valuable for them after completing treatment. They collaborated and played an active role during treatment and saw it as their responsibility to make progress during and after treatment.

*[I became active] instead of just talking about what had happened. I felt that I actually progressed in my treatment [and in life] and that I didn't just sit and repeat myself and talk about the bad things. And it was really good to actually receive concrete help. Moreover, something I could use in the future. Cause that's something I've never experienced. (F1)*

Therapy was about learning to master ailments and not about being cured, thereby helping patients accept life. The therapy had positively influenced how the patients managed everyday life in the period after treatment, and they felt that their self-image and self-confidence had improved. Some patients had already managed to return to work or study, while others needed a little more time to cope with everyday life. Others were able to pass several exams after treatment. In addition, they felt treatment had helped improve close relationships and balance the strains of life better.

*I've survived [in the past] [. . .] but I constantly sought my partner's support then [. . .] and that was a poor strategy, to manage my anxiety [without] doing it on my own. To constantly seek confirmation and support [. . .] enhances anxiety over time [. . .] I feel [the relationship] has improved [after the treatment]. And I think this is due to not having to seek confirmation and support so often now. Because I think that she found it was a bit of a drag at times, that she, in a way, had to be a support person, which she isn't trained to be. (M4)*

Commitment and willpower had been crucial for the patients' continued progress after treatment, and they were surprised and motivated by how quickly the treatment had led to improvement. Although it had been helpful, life was not without challenges. They said that learning to use the tools helped prevent deterioration after treatment ended. These patients took further steps to deal with their situation, realizing they had to do most of the job themselves to get better. As a result, they had fewer negative thoughts, and they could envisage a brighter future even though they feared relapse. They also expressed hope that they would not need treatment again.

These patients reported suffering moderately from affective disorders, anxiety disorders, or both. Their mental problems had started recently. They viewed themselves as resourceful and able to learn from therapists during brief therapy and considered this important for the therapy to work. Taking part in their treatment process gave them a feeling of coping with difficulties in life. A safe social network including work, studies, and friends made it easier to practice and use the tools they had learned during the treatment.

Patients' experiences corresponded with GPs' descriptions, suggesting brief therapy was suitable for resourceful patients with less severe symptoms, leading to improved functioning in daily life. Quick and adequate help was beneficial, and patients should not receive treatment longer than needed. In their experience, patients who were motivated for change and able to acquire what they had learned during brief therapy often had lasting effects. Such patients were relatively well-functioning beforehand, and their mental health challenges were specific, relatively recent, and moderate.

> *Brief therapy doesn't suit everyone. It must be mild to moderate depression, or less serious anxiety disorders. [. . .] Bipolar disorder or psychosis, severe depression, etc., can be difficult to manage in the brief therapy unit. (GP6)*

However, GPs also emphasized that brief therapy sometimes had a limited and temporary effect and that long-term effects depended on whether patients managed to maintain techniques they learned during brief therapy. For example, some patients discontinued cognitive exercises when they improved, sometimes resulting in relapse. Therefore, some GPs pointed out that relapse was not uncommon and could have severe consequences. Therefore, they recommended that the DPC, in the period after brief therapy, should provide some follow-up to prevent potential relapse.

> *My overall experience is that those patients who have received brief therapy relapse, after an initial period of improvement. Minor changes in their life situation make them vulnerable [. . .] They often return to their GP in the event of relapse or deterioration, and they should probably receive ongoing preventive follow-up treatment [at the DPC] while they are feeling better (GP4)*

### *Story B;* **A path to another treatment**

These patients experienced brief therapy as a comprehensive assessment process and a pathway to more suitable treatment. The main benefit of brief therapy was feeling well cared for and then being referred for more suitable treatment. However, after brief therapy ended, they were put on a new waiting list for what they described as "proper treatment", making brief therapy feel like a detour before receiving treatment adapted to meet their needs.

> *I guess I had somewhat different needs than what they could provide at the short-term clinic [. . .] I assume there were more long-term follow up needs [. . .] there was a lot of the [long-lasting depression] that I had been working on for a while now. (M7)*

They considered brief therapy not to be what they needed, and they would have preferred receiving a different treatment approach from the start. Instead of receiving treatment adapted to their needs, patients felt the therapist unsuccessfully tried to adapt them to fit the brief therapy approach.

*Well, I think that the assessment [as part of the start of brief therapy] was not so good [. . .] I felt more that the assessment . . . that the therapist wanted, in a way, to point me into the direction that my problems were thoughts, and therefore [their] form of treatment would work for me. It was more to get me to fit into [their] treatment course rather than actually assess the problems I struggle with. (M2)*

These patients experienced that brief therapy appeared to be a comprehensive assessment process that took too much time and became an unnecessary burden. Too much time had been spent mapping, assessing, or negotiating with the therapist, and they felt that they had to wait too long to get further help and adapt to a new therapist. As a result, they felt that brief therapy was a waste of time and that they should have received a more thorough assessment with several follow-up opportunities in the beginning. In addition, some patients felt that the conclusions based on brief therapy were not considered after brief therapy ended, resulting in non-coherent treatment at the same DPC.

The diagnoses these patients reported were PTSD, severe affective- or anxiety disorder, and obsessive-compulsive or bipolar disorder. In addition, some told about recurrent or complex problems like recurrent depression, severe health anxiety or phobias, and unresolved psycho-somatic conditions. Some had experienced a personal crisis prior to brief therapy. One of the GPs said:

*Brief therapy has a documented effect, and I beleive in this treatment. The problem is that the treatment is now used for everything, e.g. for manic depression or patients who have attempted to commit suicide. This doesn't make sense to me. (GP5)*

The GPs said that brief therapy served as an assessment period for some patients and agreed with patients that offering brief therapy as a form of assessment in such cases led to prolonged patient courses that did not benefit the patients and led to unnecessary use of health resources. GPs considered that brief therapy could not replace standard outpatient treatment. They reported that patients for whom brief therapy was unsuitable had nevertheless finished the full ten sessions. GPs knew that this would be insufficient, and regular outpatient treatment should have been offered initially. In addition, these patients were referred to another outpatient clinic at the DPC after brief therapy ended, involving a new waiting period and a new therapist, leading to patients' frustration and impatience. Another GP said:

*If brief therapy becomes a kind of examination, we could think of it as part of a longer treatment course, rather than an unsuccessful treatment. If the event that the treatment was not the right one for the patient, they nevertheless have entered "the [treatment] system" (GP1)*

## Story C; Confusion and lost hope

These patients reported that brief therapy was inappropriate due to complex challenges. In the beginning, they relied on the therapist's expertise, feeling that progress depended on them and believing that the treatment would work. Therapists and patients collaborated at the start of the treatment to develop a treatment plan to achieve specific goals within a limited time. They

perceived this plan as limited by the therapist's focus and did not address the complexity of their problems. Some were upset when they experienced not getting better and did not understand the brief therapy exercises. They were unable to manage the treatment plan, which they considered their responsibility. They felt they had been given too much responsibility during and after therapy. These patients felt a sense of failure when they did not understand or carry out the exercises, thereby not meeting the therapists' expectations. Due to fear of losing the treatment offer, some did not shar these thoughts with the therapist.

> *Knowing that when I'm in [the treatment program], then it's important to kind of keep the place I've been given, until I improve [. . .] Yes, so it's hard to tell, kind of that–now it's not going so well. I was afraid that I would lose [the offer], sort of being on my own again. (F3)*

These patients described feeling confused when treatment started and unable to understand the cause of their ailments, which they thought the "expert therapists" would assess. They were disappointed not to receive the help they needed. Some of these patients found the therapist's focus and questions during treatment sessions made it difficult to tell them that they had other problems besides anxiety or depression, feeling there was no room to consider other diagnoses or mental health challenges. For them, there was insufficient time to assess or negotiate what they should work on before the actual treatment started. Due to severe mental health problems, they had expected to receive something different and more comprehensive than brief therapy. When they did not experience any improvement after brief therapy and did not receive any other treatment, they felt confused and lost hope of recovery. They said that the therapists at the end of the program had concluded that they had more extensive problems than previously known and that the therapist gave them a new diagnosis they were unfamiliar with. After brief therapy, these patients felt worse and were unsure why they had not received more treatment within the specialist health service.

> *Well . . . yes, it's been really quite bad [. . .] to be honest. It . . . after treatment was completed, after the last session, I received a summary letter by post, where I got an entirely new diagnosis that I had not heard about before. I did not expect to be given a new diagnosis by post without any further follow-up [and it did not seem like] the GP or other doctors really understood it either. (M2)*

Receiving a new diagnosis at the end of brief therapy surprised these patients, and they felt not involved in the assessment process that resulted in the diagnosis. In addition, they had initially expected that brief therapy would be a collaborative treatment project, but the therapist just informed them that the treatment team had discussed a new diagnosis before the treatment ended. Thus, according to them, the patient was presented with a surprising conclusion at the end of brief therapy. As a result, the patients did not feel the new diagnosis was helpful but instead confusing, and it suggested that the problems were permanent and not something they would be able to manage. Patients described it as detrimental to receive what they, in retrospect, thought was an incorrect treatment. These patients reported suffering from severe mood- or anxiety disorders, personality disorders, and bipolar disorder. They received follow-up from the GP after brief therapy ended.

GPs pointed out that brief therapy was not suitable for patients with severe, complex, or recurrent mental health problems, and some patients with chronic disorders or severe ailments deteriorated after receiving brief therapy. Since there was little information about the various treatment options at the DPC, some GPs expressed referring patients to outpatient treatment without knowing which treatment the patients would receive.

*I think it is important that the referring GP knows what kind of offer and treatment scheme they are referring the patient for. Some of the patients are disappointed by the treatment, and I believe that this may be due to insufficient information. Both the referring GP and the patient should be well informed up front. Information about the purpose of the treatment should be made clear. (GP1)*

These GPs did not participate in any discussions about the assessment. GPs had experienced that the therapy had been unsuccessfully provided when patients were not adequately examined or evaluated beforehand. They noted that patients without jobs or studies were the most vulnerable. Some GPs found that patients they referred for outpatient treatment for severe problems were nevertheless offered brief therapy after the assessment at the DPC. In some of these cases, the GPs did not believe that brief therapy would work for the patients and had experienced that their concerns were validated afterwards. The GPs said it was problematic that they had no influence on which treatment the patient was offered at the DPS. In addition, some GPs felt left alone with patients who had extensive treatment needs after brief therapy ended, and these GPs wanted more guidance from the DPC on the way forward. The information provided by the DPC to the GP following brief therapy was insufficient and of little help, warranting more extensive cooperation with the DPC for follow-up of patients with extensive healthcare needs. One of the GPs said:

*The* summary *from the DPS may state that the patient benefited from the treatment, resulting in reduced anxiety, but then they return to me even more in despair because they feel that it was of no use. They were unable to establish the rapport they wished for, and were informed that it was an untreatable chronic condition. These patients felt more hopeless. (GP5)*

## Collaborative meetings with the DPC

Another finding was that the GPs who participated in collaborative meetings with the DPC said that the collaboration had contributed to discussions and a shared learning process that led to more suitable referrals for brief therapy. The GPs said that this collaboration made it easier for them to follow up with patients with extended needs, and they felt more confident in their role as GPs. GPs without such collaborative meetings said that they did not participate in any discussions regarding the offer their patients would receive. Since there was little information about the different treatment offers at the DPC, they referred patients to outpatient treatment without knowing what treatment the patients would receive. The differences can be illustrated by two quotes from the GPs:

*The DPC determines which offer may suit the patients I refer to them, following an assessment interview. I am not involved in this [process]. (GP3)*

And another GP expressed it like this:

*The process of referral for brief therapy works really well for us, because DPC admission teams come to us every two weeks. It involves a specialist and another person from the intake team who consider all referrals [. . .] An advantage to us of having an admission meeting with the DPC, is that we only need to present the issues to them and they can then suggest what kind of offers may be suitable for the patients (GP2)*

## Discussion

The initial premise of the authors was that understanding the experiences of both general practitioners (GPs) and patients could offer valuable insights when evaluating innovation and/or improvement initiatives within mental health services. While the results of this study may not be applicable to all levels of the healthcare system such as primary and specialist healthcare services, the clarity of the findings from this qualitative research might be robust enough to establish a foundation for hypothesis testing in future research about who benefits from various treatment options. In this study patients suffering moderately from affective disorders, anxiety disorders, or both reported beneficial experiences with brief therapy. It seemed that patients experienced brief therapy as beneficial if they understood the purpose of brief therapy and felt able to actively participate in the treatment. However, for other patients brief therapy seemed to serve as an assessment period, leading to prolonged patient courses and unnecessary use of health resources. Brief therapy was described as unsuitable for patients with severe, complex, or recurrent mental health problems, and some of the patients with self-reported chronic disorders and severe problems even experienced deterioration after receiving brief therapy. Both patients and GPs noted that inadequate referrals and assessments occasionally resulted in more severely affected patients being offered brief therapy. Nevertheless, collaborative meetings between the GPs and the DPC seemed to contribute to discussions and a shared learning process, leading to more suitable referrals of patients to brief therapy.

### Was brief therapy experienced as suitable for those who received it?

Both patients and GPs described brief therapy as suitable for relatively well-functioning patients with mental problems that recently started. Our results align with research showing that brief therapies are efficient for common mental disorders [9,27,28] and that brief therapy seems to provide good treatment outcomes for patients suffering from moderate depression, anxiety, or both [20]. However, the study showed that brief therapy appeared as provided to patients with more severe and complex mental health problems, and this group of patients benefited less from brief therapy. Providing the appropriate treatment adapted to meet the individual patient's needs is essential for the whole health service. However, this goal has been challenging to achieve because of limited knowledge about the suitability of different lengths and modes of therapies [29].

In this study, brief therapy seemed to improve everyday life with work, studies, and close relationships among patients who experienced the treatment as beneficial. These were the patients who were able to learn the skills in therapy to cope with thoughts, feelings, and events and anticipate when and how they could apply the skills learned in therapy for future situations [30]. A possible explanation for why these patients improved was that the treatment reduced fear and pessimism, thereby helping the patients cope and function better in daily life [28,31]. Therefore, the functional capacity for long-term maintenance may depend on the patient's level of education, the work situation, the social network, and the severity of depressive symptoms before treatment and might be a predictor for the suitability of different types of treatment [29,30].

Our findings indicate that patients experienced brief therapy as beneficial when they understood the purpose of brief therapy and felt able to actively participate in the treatment, and their possibility of agreement on goals seemed to depend on this ability. Meta-analyses have demonstrated links between goal consensus, collaboration, and patient outcomes in individual psychotherapy [32], and the researchers recommend only beginning work on patients' mental health problems after establishing a goal consensus [32]. It is argued that patients' suitability for treatment should be assessed against the extent to which patients' problems have an

identifiable focus (nature of problems), whether the patient has the ability for flexible interaction (ego strength), has reflective abilities (self-observing capacity), and also motivation for the specific psychotherapeutic treatment [29]. Our findings highlight the importance of limiting the offer of brief therapy to the defined target group of patients with moderate ailments [31] who have the opportunity and ability to participate in working with defined issues.

## The importance of proper assessment before referral

In the present study, some GPs had participated in collaborative meetings with the DPC, while others had not. Communication with the DPC in the referral process influenced GPs' experiences of whether patients were provided adequate treatment in the DPC. This was affected by whether GPs and patients had the opportunity to provide valuable input to the DPC's choice of treatment approach. Nevertheless, most patients did not know what kind of treatment they would receive before meeting the therapist, thus only participating after the treatment sessions started. The involvement of patients is considered crucial to promote health and ensure the quality of health services [33]. Therefore, before starting treatment, the patient's preferences and desires for involvement should be assessed, and how these could be met should be considered [33].

Since health care in Norway is organized at two levels–primary and secondary health services–coordination of mental health services to meet an individual's needs across these different care levels is essential for ensuring quality care for people with severe mental health ailments. [34]. Our findings, related to the importance of collaboration meetings, align with studies that show a need for better communication between a patient's different health providers and relevant patient information between therapists and GPs when choosing suitable treatment [35]. In addition, effective coordination seems crucial for the perceived treatment quality for people with severe mental illness who move between care settings [34]. People with more complex mental health problems seem to require multiple interventions and follow-up from multiple services [36]. Therefore, systems and procedures should be developed to ensure clear responsibility and transparency at every stage of the treatment course [37]. In addition, research has shown that professionals have mixed success in identifying and managing patients' needs in transitions between services [37], which should keep the focus on moving forwards.

Overcoming the challenges of coordinating continuity in care is crucial for people with complex mental health challenges [34], and communication across professions seems essential to benefit patients with moderate to severe ailments [38]. However, the GPs, particularly those without regular collaboration meetings, perceived the treatment options for patients as fragmented and poorly communicated. In addition, they felt "left alone" with patients who did not benefit from brief therapy, and they missed adequate follow-up treatment from the specialist health services. Strengthening collaboration between the DPC and GP offices in the catchment area, for instance, through collaboration meetings, would be helpful. Communications routines between the health providers might help ensure that brief therapy is provided to the patients benefiting from such treatment. In addition, more coherent treatment, and collaboration on follow-up over time can constitute a more binding agreement for the follow-up of patients with more extensive needs.

## Experiences of misuse of resources and possible poor treatment

In the present study, some patients with severe or recurrent mental health problems received additional and adequate treatment after ending brief therapy. For them, the brief therapy treatment extended the waiting time and became a detour to appropriate treatment, neither

experienced as benefiting the patients nor making good use of health resources. Here, brief therapy seemed to serve as an assessment period and became a pathway to more suitable treatment. Unfortunately, after brief therapy ended, the referral to another outpatient clinic at DPC implied extra waiting time and a new therapist, leading to patients' frustration and impatience, making brief therapy feel like a detour before receiving treatment adapted to their needs. Waiting times affect treatment outcomes, and research shows that outcomes are worse when waiting times are longer [16]. Caring for people with complex ailments warrants a complementary strategy, supporting GPs and therapists to offer personal, comprehensive continuity in care [39]. When brief therapy in some cases was experienced as inappropriate, the treatment resources in brief therapy appeared as a long-term assessment period and not a treatment. Effective organization of available mental health services should ensure adequate distribution of specialized mental health services [40]. As part of effective organization, brief therapy should not be used as an assessment period to provide adequate and effective treatment.

The most disturbing finding in this study was that some patients with self-reported severe ailments experienced that they did not receive adequate treatment after ending brief therapy. They expressed a loss of hope for recovery and felt that their mental health deteriorated. Although essential contributions have been made in recent years to help adults with anxiety or depression, research calls for a focus on comorbid mental and physical conditions that must also be addressed [16]. This topic should be followed up in future research.

The severity of the patient's condition is probably the most critical factor in GPs' decisions to refer patients to the DPC [40], and evaluating individual needs seems essential. However, GPs without collaboration meetings experienced missing guidance and cooperation on follow-up of patients, both before and after treatment. Therefore, interventions such as brief therapy should address the patient's health outcomes, in the sense that clinical processes should be developed to improve practice and collaboration between health services [41]. Patients should be involved in evaluating brief therapies [42] to ensure a sense of "ownership" of their treatment process and prevent loss of hope and deterioration. Loss of hope can decrease patients' motivation to be involved in their recovery processes [43]. It is possible that patients may perceive a refusal for further treatment following brief therapy as a refusal from the entire specialist health service. Choosing the optimal treatment strategy for a particular patient can help avoid unnecessary suffering and treatment costs. The probable outcomes of different types of treatments should therefore be carefully discussed in advance [44]. Thus, our findings support an increased clinical focus on giving sufficient time to cooperate and assess the patient's needs before determining the type of treatment. This study also shows that there is a need for more knowledge about who benefits from various treatment options, so that healthcare interventions for mental health problems can be better tailored to meet the individual patient's needs.

## Strengths and limitations

The sample consisted of both patients and GPs and variation in age and gender. Another strength that helped shed light on the patients' process after the end of treatment was the extended time- period following brief therapy (minimum 12 months), providing patients with an opportunity to reflect on long-term experiences. In addition, the GP sample included diversity in experiences of referring several patients to the DPC since 2016 and whether the GPs had attended collaboration-meetings with the DPC or not. Due to the Norwegian socio-cultural context, such as extensive mental health and welfare services, transferability to other countries may be limited. All patients in this study completed brief therapy between 1 and 8 months before the worldwide COVID-19 pandemic in 2020. We collected data in the middle of a pandemic that may have affected mental health at the individual and societal levels. A

limitation of this study might be that informants' reports on potential benefits after receiving brief therapy may be affected by this situation.

A strength of this study was the richness and complexity of the patients' personal stories. The validity of the narratives was strengthened through the transparent process used to form and describe condensed stories [26]. Including the experiences of both patients and GPs provided a more diverse description of experiences with brief therapy. The inclusion of GPs from different clinics contributed to a broader range of descriptions of collaboration with the DPC. Even though GPs were limited in number, each GP had extensive experience referring patients to this DPC.

The authors' complementary experiences strengthened the analysis process by providing different professional perspectives. In addition, discussions with researchers in another research group strengthen the study by providing alternative considerations.

A limitation of this study may be that patients 'and GPs' experiences are not confirmed by other data types. Also, this qualitative study cannot measure effect of treatment. There is a need for studies evaluating who benefits from what type of treatment. Furthermore, the generalizability of the results is uncertain since data was collected from a limited number of GPs and patients within the context of a single DPC. However, the findings may be considered in other DPCs and might provide a starting point for further research investigating the patients' long-term outcomes after brief therapy and to follow-up with patients with more comprehensive needs.

## Conclusion

This study provides insight into patients' experiences and reflections one year after receiving brief therapy, as well as GPs' experiences after referring several patients to a district psychiatric centre (DPC) who have received brief therapy since 2016.The findings are based on patients' and GPs' experiences with brief therapy, as defined in this study. According to both patients and GPs, brief therapy was found to be beneficial for individuals experiencing moderate affective disorders, anxiety disorders, or both. It was experienced to be unsuitable for individuals with more severe, complex, or recurrent mental health conditions. Nevertheless, both patients and GPs thought that insufficient referral-procedures had led to some patients who were more severely affected by mental disorders being offered brief therapy. It seems imperative to examine the consequences of possible insufficient referral routines both for the individual patient and for the use of the health service's resources. There also seems to be a need for more knowledge about who benefits from various treatment options, so that healthcare interventions for mental health problems can be better tailored to meet the individual patient's needs. In this study, collaborative meetings between the GPs and the therapists at DPC was experienced to facilitate more suitable referrals of patients to brief therapy. Therefore, implementing an effective system for providing relevant information between patients, GPs, and the DPC could help manage expectations, facilitate two-way communication, and potentially lead to both appropriate referrals and -assessments. This includes ensuring that patients, GPs, and therapists are well-informed about assessment procedures, treatment options, expectations, and follow-up strategies. Such an approach may potentially increase the likelihood of successful treatment outcomes.

## Supporting information

**S1 File.**
(DOC)

## Acknowledgments

We thank all the patients and GPs for taking part in the interviews. We also thank the Competence Center for Lived Experience and Service Development representatives in Central Norway for helpful input to the research project.

## Author Contributions

**Conceptualization:** Hilde V. Markussen, Marit B. Rise.

**Data curation:** Hilde V. Markussen, Marit B. Rise.

**Formal analysis:** Hilde V. Markussen, Lene Aasdahl, Petter Viksveen, Marit B. Rise.

**Funding acquisition:** Hilde V. Markussen, Marit B. Rise.

**Investigation:** Hilde V. Markussen, Marit B. Rise.

**Methodology:** Hilde V. Markussen, Marit B. Rise.

**Project administration:** Hilde V. Markussen.

**Resources:** Hilde V. Markussen.

**Software:** Hilde V. Markussen.

**Supervision:** Lene Aasdahl, Petter Viksveen, Marit B. Rise.

**Validation:** Hilde V. Markussen, Lene Aasdahl, Petter Viksveen, Marit B. Rise.

**Visualization:** Hilde V. Markussen, Petter Viksveen.

**Writing – original draft:** Hilde V. Markussen.

**Writing – review & editing:** Hilde V. Markussen, Lene Aasdahl, Petter Viksveen, Marit B. Rise.

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
