## [Decision Letter · Decision Letter 0]

4 Apr 2024

PONE-D-24-03765Who benefits from brief therapy for mental health problems in the long term? An exploration of patients' and general practitioners' experiences: A Qualitative StudyPLOS ONE

Dear Dr. Markussen,

Thank you for submitting your manuscript to PLOS ONE. After careful consideration, we feel that it has merit but does not fully meet PLOS ONE’s publication criteria as it currently stands. Therefore, we invite you to submit a revised version of the manuscript that addresses the points raised during the review process.

We look forward to receiving your revised manuscript.

Kind regards,

Ramune Jacobsen

Academic Editor

PLOS ONE

Journal Requirements:

Reviewers' comments:

Reviewer's Responses to Questions

**Comments to the Author**

1. Is the manuscript technically sound, and do the data support the conclusions?

Reviewer #1: No

Reviewer #2: Partly

2. Has the statistical analysis been performed appropriately and rigorously? 

Reviewer #1: N/A

Reviewer #2: N/A

3. Have the authors made all data underlying the findings in their manuscript fully available?

Reviewer #1: No

Reviewer #2: No

4. Is the manuscript presented in an intelligible fashion and written in standard English?

Reviewer #1: Yes

Reviewer #2: Yes

5. Review Comments to the Author

Reviewer #1: This qualitative study explores the experiences of patients and GPs about referral for short term mental health therapy.

There are broadly three sections to this paper, an introduction, the actual study, and finally a discussion/conclusion. The main problem with the paper is that the study itself is not well framed as neither the introduction nor conclusion have a clear relationship with the main analysis.

The introduction largely covers the effectiveness of brief therapy for mental health problems, in this case CBT, and proposes to ‘determine who experience benefits from brief therapy in the long term’. Indeed, the title of the paper is ‘who benefits …’. But there is no way that a study of 11 patients can determine who benefits from treatment; that would require a trial involving hundreds of patients. We know already from lots of such studies that brief CBT does benefit patients with mild depression/anxiety – though not everyone. That is where this study can contribute by suggesting types of referral where benefit may be low – though it cannot identify those patients with any degree of certainty. Essentially, this is a qualitative study which is about hypothesis-generation and not about hypothesis-testing.

The discussion largely addresses policy and a number of ways in which treatment ‘should’ be delivered. But these conclusions are not supported by the empirical material presented here. Asserting, for example, that ‘brief therapy should address the patient's health outcomes, in the sense that clinical processes should be developed to improve practice and collaboration between health services’ or that ‘patients should be involved in evaluating brief therapies to ensure a sense of "ownership" of their treatment process and prevent loss of hope and deterioration’ cannot be inferred from any findings from this study.

The value of this study therefore is that it begins to explore the sorts of patients who may or may not benefit from brief CBT. Of course, we do not know whether or not these patients interviewed were judged as ‘recovered’ (by some psychiatric assessment) at the end of treatment but it is still useful to have patients’ own assessment in their own words of what benefit they thought they derived from the therapy, especially a year later. The categorization of responses into three groups is helpful though perhaps might have been predicted – therapy works for some patients, and it doesn't work for others, some of whom present fairly intractable problems and were perhaps inappropriately referred. Nevertheless, these are interesting findings. The ‘evidence’ presented in support, however, is fairly skimpy (two quotations per type) and we have to assume that all 11 patients fell neatly into one of the three groups without any anomalous findings.

Management of the GP interview data (six interviews) is a bit more opaque. The GPs’ views are simply used to ‘challenge or elaborate on relevant aspects’ of the three types of patients revealed by analysis of the patient data. How did GP experiences map on to patients’ experiences? They have different perspectives yet, according to this analysis’ their experience was congruent with that of the patient. It is not clear, for example, whether GPs would have come up with the same three types of patients if their data had been analysed independently. The ‘GP voice’ also seemed to be missing. According to this analysis GPs agreed that some patients were inappropriately referred – but these are the people doing the referring. How do they explain making ‘inappropriate’ referrals (or is it just ‘other’ GPs doing that)? Further, some patients with mild anxiety/depression can be very demanding and referral might not only be used to potentially help the patient but also to give some respite for the GP. But maybe these GPs did not mention these aspects of ‘their’ experience?

In summary, there is an interesting core to this paper but its framing in terms of effectiveness and policy is misplaced given that it contributes very little to either area. What the authors could do is address the existing literature on patients’ experiences of medical treatment and how these might differ from the formal assessments that are likely to be given at the end of therapy to see whether or not the patient has recovered. Then perhaps some more about the study proper with additional material from the interviews, and a conclusion that shows what hypothesis have been generated from the analysis.

Reviewer #2: Thank you for letting me review this interesting manuscript of a very important subject. I think the work is ambitious and to most parts well described with reasonable conclusions. I would recommend the authors to attach a Checklist of qualitative studies such as the COREQ, because there are some questions there which could be more clearly described, e.g. the relationships of the researchers to the participating, patients, GPs and to the project?

Introduction:

It was a bit hard to find out what the definition of “brief therapy” is. In Primary Care it could mean 3-4 sessions, but here it is unclear, in p13:90 4-26 sessions are mentioned? On p 12:82-84 there is also no difference mentioned between what is considered brief PDT compared to CBT, in the latter 12 sessions are considered standard, not brief.

Later in the manuscript the included patients received 8-15 sessions of either CBT or MCT, which is standard for these treatments not brief. Kind of therapy and if diagnose-specific or unified protocols were used? Needs clarification.

Methods:

The diagnostic procedure seems lacking, including severity assessment, which the authors also conclude could be a major cause of the findings. Also makes me wonder about the design of the project, e.g. treatment for PTSD is specific CBT, prolonged exposure (PE) according to guidelines and research evidence, why give these patients some brief intervention based on any diagnosis of anxiety or depression? One size fits all?

I guess it relates to the previous study but to be able to interpret the qualitative interviews it is important to know: When and by whom were the assessments made, which kind of treatment protocols were used? Inclusion, exclusion criteria? Pharmaceutical treatment? Protocol if patient got worse, suicidal? Because there seems to be a mix of conditions, these factors are important to interpret the results.

Qualitative methods:

P14:133: Recruitment and selection of participants: Out of 459 treated patients 11 participated in the interviews. How many were invited? How did the authors secure a trustworthy generalizable sample? Was there a defined saturation point of variability from the interviews?

P14:139; P15:145: Selected by purposeful criterion sampling, please clarify, what criteria?

Procedure, analysis: The interviews and processes are well described but the thematic analysis and coding process remains unclear. Are the three different “stories” categories or themes? Subcategories?

P18:220 description of meaning units and coding in groups more similar to content analysis? To me the analysis process remains unclear.

Results:

Table 2: Almost seems like a quantitative analysis and report of how many patients were in the three groups (“stories”). In that case we lack the power of a quantative analysis with just 11 participants with very different kind of conditions and experiences of previous care according to table 1. Makes it hard to ensure transferability and generalizability.

Good and clarifying to see quotations of each story, but to ensure trustworthiness of the coding process a table or some example of coding the meaning units would clarify.

Limitations:

As described, pre-understanding can be positive but could also affect the interpretation and analysis if not made aware. Please describe researchers relationship to the material, and clarify how the close-to-text-analysis was performed.

As mentioned above there are several limitations to the recruitment and analysis which means difficulties to generalize the findings to all patients and GPs in the project, but also to other health care in general.

Conclusions:

The conclusions seem adequate and obvious in this context, but if “brief intervention” can stand for a mix of interventions (CBT, MCT) of 8-15 sessions and for different conditions, with different complexity and severity it is hard to draw any further conclusion of the intervention. The participants in story A maybe could have been helped with an even briefer intervention or maybe a group or an internet mediated intervention, whereas we know that people with bipolar, PTSD or other severe conditions need specific interventions.

But this is important knowledge if not known before. P32:572 assessment of main diagnosis, comorbidity and severity is necessary to refer patients to correct level and kind of care.

6. PLOS authors have the option to publish the peer review history of their article (what does this mean?). If published, this will include your full peer review and any attached files.

Reviewer #1: **Yes: **David Armstrong

Reviewer #2: No

---

## [Author Response · Author response to Decision Letter 0]

16 Jun 2024

PLOS ONE

Academic Editor

Ramune Jacobsen

Response to reviewers from Hilde V Markussen et.al - June 2024

PONE-D-24-03765

Perceived benefits and challenges one year after receiving brief therapy in a district psychiatric centre. An exploration of patients' and GPs' experiences: A qualitative study (NB: The title is revised and changed)

Dear Editor,

We gratefully acknowledge the invitation to submit a revised version of the manuscript. We have addressed and incorporated the helpful suggestions from the reviewers and responded to their comments. Please find our responses to their comments below.

Authors’ reply:

We have updated our ethics statement. We have removed other information about ethics from the manuscript (Page 15, l. 324-327). 

This study was approved by the “Regional Committee for Medical and Health Research Ethics” (REK) in central Norway (2018/49). REK approved the information letter before we used it in the study. Email: rek-midt@mh.ntnu.no

Reviewers' comments, Reviewer's Responses to Questions, Comments to the Author:

1. Is the manuscript technically sound, and do the data support the conclusions?

Reviewer #1: No

Reviewer #2: Partly

Authors' response: 

The manuscript describes a qualitative study, and we have worked to improve the text to make it clear that the data support the discussions and conclusions of our study.

In order to elaborate on this, we have rephrased parts of the text, including the abstract (Page 3-4) 

3. Have the authors made all data underlying the findings in their manuscript fully available?

Reviewer #1: No

Reviewer #2: No

Authors' response: 

Data availability is now described on page 34 (l. 790-796). 

There are both ethical and legal restrictions on sharing the data set due to sensitive information. It is not possible to publish the original data as participants were guaranteed that their interviews would not be made publicly available. Therefore, data publication would violate their privacy rights and conflict with the General Data Protection Regulation (GDPR). The approved information letter to participants in the present study stated; That the researchers have a duty of confidentiality, and information from the interviews will not be given to employees at the DPC nor anyone else.

5. Review Comments to the Author

Reviewer #1

Reviewer #1: This qualitative study explores the experiences of patients and GPs about referral for short term mental health therapy.

Authors' response: 

Thank you for the remark. To clarify, our study aimed to explore the experiences of patients who had undergone brief therapy at a district psychiatric centre a minimum of one year ago and the experiences of general practitioners (GPs) who had referred patients to such treatment since 2016, in order to explore perceived benefits from brief therapy in the long term. We have changed the manuscript's title so that the title matches the manuscript to a greater extent, as the reviewer points out.

Our title was as follows: Who benefits from brief therapy for mental health problems in the long term? An exploration of patients' and general practitioners' experiences: A Qualitative Study. Our revised/NEW title is as follows: Perceived benefits and challenges one year after receiving brief therapy in a district psychiatric center. An exploration of patients' and GPs' experiences: A qualitative study

Reviewer #1: There are broadly three sections to this paper, an introduction, the actual study, and finally a discussion/conclusion. The main problem with the paper is that the study itself is not well framed as neither the introduction nor conclusion have a clear relationship with the main analysis.

Authors' response: 

We have rewritten the conclusion to link it more closely to the analysis (page 32-33).

Reviewer #1: The introduction largely covers the effectiveness of brief therapy for mental health problems, in this case CBT, and proposes to ‘determine who experience benefits from brief therapy in the long term’. Indeed, the title of the paper is ‘who benefits …’. But there is no way that a study of 11 patients can determine who benefits from treatment; that would require a trial involving hundreds of patients. We know already from lots of such studies that brief CBT does benefit patients with mild depression/anxiety – though not everyone. That is where this study can contribute by suggesting types of referral where benefit may be low – though it cannot identify those patients with any degree of certainty. Essentially, this is a qualitative study which is about hypothesis-generation and not about hypothesis-testing.

Authors' response: 

Thank you for this remark. We have added text in the manuscript to clarify that this study does not aim to determine any benefit of the intervention (Page 6-7). 

We have also changed the manuscript's title to clarify this.

Reviewer #1: The discussion largely addresses policy and a number of ways in which treatment ‘should’ be delivered. But these conclusions are not supported by the empirical material presented here. Asserting, for example, that ‘brief therapy should address the patient's health outcomes, in the sense that clinical processes should be developed to improve practice and collaboration between health services’ or that ‘patients should be involved in evaluating brief therapies to ensure a sense of "ownership" of their treatment process and prevent loss of hope and deterioration’ cannot be inferred from any findings from this study.

Authors' response: 

Many thanks for this comment. To clarify, the findings of this study were analyzed in the context of politics and fields of practice, with the aim of generating hypothesis that could be further developed and tested in subsequent studies. We have revised the text accordingly in the manuscript on page 26, l. 601-603:

Both patients and GPs noted that inadequate referrals and assessments occasionally resulted in more severely affected patients being offered brief therapy.

And, page 28: Since health care in Norway is organized at two levels – primary and secondary health services – coordination of mental health services to meet an individual's needs across these different care levels is essential for ensuring quality care for people with severe mental health ailments. [34]. Our findings, related to the importance of collaboration meetings, align with studies that show a need for better communication between a patient’s different health providers and relevant patient information between therapists and GPs when choosing suitable treatment [35]. 

Reviewer #1: The value of this study therefore is that it begins to explore the sorts of patients who may or may not benefit from brief CBT. Of course, we do not know whether or not these patients interviewed were judged as ‘recovered’ (by some psychiatric assessment) at the end of treatment but it is still useful to have patients’ own assessment in their own words of what benefit they thought they derived from the therapy, especially a year later. The categorization of responses into three groups is helpful though perhaps might have been predicted – therapy works for some patients, and it doesn't work for others, some of whom present fairly intractable problems and were perhaps inappropriately referred. Nevertheless, these are interesting findings. The ‘evidence’ presented in support, however, is fairly skimpy (two quotations per type) and we have to assume that all 11 patients fell neatly into one of the three groups without any anomalous findings.

Authors' response: 

Thank you for pointing this out. We have made several changes in the manuscript to clarify that the study cannot determine which type of patients who may or may not benefit from brief therapy (The title and abstract, page 3 and 4)

A title under the discussion-section is also rewritten, from “For whom is brief therapy experienced as suitable” to: Was brief therapy experienced as suitable for those who received it? (Page 26)

Under the Methods-section, we have added some sentences to clarify the approach (page 7)

Under “Main results” we have added a sentence (Page 17): Some patients are represented in more than one condensed story and […]

The non-verbatim presentations of the participants‘ experiences became more generally representative accounts of the participants’ experiences (and not the presentation of "evidence"). There are, of course, many quotes that could be used, but for reasons of space, more have not been included. If it is desired we are happy to add more quotations.

Reviewer #1: Management of the GP interview data (six interviews) is a bit more opaque. The GPs’ views are simply used to ‘challenge or elaborate on relevant aspects’ of the three types of patients revealed by analysis of the patient data. How did GP experiences map on to patients’ experiences? They have different perspectives yet, according to this analysis’ their experience was congruent with that of the patient. It is not clear, for example, whether GPs would have come up with the same three types of patients if their data had been analysed independently. The ‘GP voice’ also seemed to be missing. According to this analysis GPs agreed that some patients were inappropriately referred – but these are the people doing the referring. How do they explain making ‘inappropriate’ referrals (or is it just ‘other’ GPs doing that)? Further, some patients with mild anxiety/depression can be very demanding and referral might not only be used to potentially help the patient but also to give some respite for the GP. But maybe these GPs did not mention these aspects of ‘their’ experience?

Authors' response: 

Thank you. We have tried to further clarify this. The GPs did not mention the aspects referred to here, and it was therefore not part of our findings/data.

The GPs did not assess the referrals to offer treatment at DPC. We have added text in the manuscript to elaborate on the process of referrals (Page 8) Patients were mainly referred to the DPC from their GP, or by psychologists or psychiatrists in private practice. The DPC assessed the referrals and decided whether treatment was needed or not and offered a right to treatment within a specific time period (a waiting period guarantee).

We have also rewritten sentences and added text in the manuscript at page 21.

We have also incorporated a few quotes from GPs into the text to illustrate the findings.

Reviewer #1: In summary, there is an interesting core to this paper but its framing in terms of effectiveness and policy is misplaced given that it contributes very little to either area. What the authors could do is address the existing literature on patients’ experiences of medical treatment and how these might differ from the formal assessments that are likely to be given at the end of therapy to see whether or not the patient has recovered. Then perhaps some more about the study proper with additional material from the interviews, and a conclusion that shows what hypothesis have been generated from the analysis.

Authors' response: 

Thank you for pointing out that there is an interesting core to the paper. We have provided further insights into the potential importance for policy and practice. This article is one out of three in a coherent research project. In the overall project, and in this article's framework; the participants’ perspectives of the benefits of the intervention and the health policy are important for understanding different challenges faced by patients, GPs and therapists.

We have added text to the manuscript to elaborate on this (Page 8).

Reviewer #2 

I would recommend the authors to attach a Checklist of qualitative studies such as the COREQ, because there are some questions there which could be more clearly described, e.g. the relationships of the researchers to the participating, patients, GPs and to the project?

Authors' response:

Thank you. We have added a COREQ checklist as appendix to the manuscript.

Reviewer #2: Introduction:

It was a bit hard to find out what the definition of “brief therapy” is. In Primary Care it could mean 3-4 sessions, but here it is unclear, in p13:90 4-26 sessions are mentioned? On p 12:82-84 there is also no difference mentioned between what is considered brief PDT compared to CBT, in the latter 12 sessions are considered standard, not brief.

Later in the manuscript the included patients received 8-15 sessions of either CBT or MCT, which is standard for these treatments not brief. Kind of therapy and if diagnose-specific or unified protocols were used? Needs clarification.

Authors' response:

We agree and have added “Brief therapy” under “List of abbreviations” (Page 32): Brief therapy: At this DPC, brief therapy was described as time-limited cognitive psychotherapy restricted to ten treatment sessions and included CBT and metacognitive therapy and was offered to adult patients over 18 years of age in general, with moderate to severe anxiety, depression, and other mental disorders.

In the introduction, we inform that the terms "brief therapy", "time-limited therapy "and "short-term therapy" are used interchangeably in the literature. We have added a sentence to clarify (Page 5).

In the study setting-section (page 7) we have added some words and sentences to clarify how brief therapy was described in this setting/this study:

[…] At this DPC, brief therapy was described as time-limited cognitive psychotherapy restricted to ten treatment sessions [7]. According to the head of the brief therapy unit, brief therapy included CBT and metacognitive therapy [21] and was offered to adult patients over 18 years of age in general, with moderate to severe anxiety, depression, and other mental disorders. The head of the unit also described the time-limited treatment as more flexible than 10 treatment hours. Depending on the individual patient’s needs, it could be shorter or extended by a few ours […] The DPC leadership described that brief therapy was established to provide metacognitive and cognitive behavioral therapy to patients with anxiety and depression as the leading cause of referral […]

Reviewer #2: Methods:

The diagnostic procedure seems lacking, including severity assessment, which the authors also conclude could be a major cause of the findings. Also makes me wonder about the design of the project, e.g. treatment for PTSD is specific CBT, prolonged exposure (PE) according to guidelines and research evidence, why give these patients some brief intervention based on any diagnosis of anxiety or depression? One size fits all?

I guess it relates to the previous study but to be able to interpret the qualitative interviews it is important to know: When and by whom were the assessments made, which kind of treatment protocols were used? Inclusion, exclusion criteria? Pharmaceutical treatment? Protocol if patient got worse, suicidal? Because there seems to be a mix of conditions, these factors are important to interpret the results.

Authors' response:

We see the reviewer’s point. In this research project we have reported on the experiences of patients and GPs concerning all categories of patients who received brief therapy at the DPS. The results suggest that patients with more severe and persistent mental health problems seemed to respond less positively (or not at all) to receving brief therapy. Hence, the results suggest that a differentiation between categories of patients depending on the type, severity and persistence of mental health problem should be carefully considered prior to offering the treatment for them. This would be

---

## [Decision Letter · Decision Letter 1]

9 Oct 2024

Perceived benefits and challenges one year after receiving brief therapy in a district psychiatric centre. An exploration of patients' and GPs' experiences: A qualitative study

PONE-D-24-03765R1

Dear Dr. Markussen,

We’re pleased to inform you that your manuscript has been judged scientifically suitable for publication and will be formally accepted for publication once it meets all outstanding technical requirements.

Kind regards,

Ramune Jacobsen

Academic Editor

PLOS ONE

Additional Editor Comments (optional):

Reviewers' comments:

Reviewer's Responses to Questions

**Comments to the Author**

1. If the authors have adequately addressed your comments raised in a previous round of review and you feel that this manuscript is now acceptable for publication, you may indicate that here to bypass the “Comments to the Author” section, enter your conflict of interest statement in the “Confidential to Editor” section, and submit your "Accept" recommendation.

Reviewer #1: (No Response)

2. Is the manuscript technically sound, and do the data support the conclusions?

Reviewer #1: Yes

3. Has the statistical analysis been performed appropriately and rigorously? 

Reviewer #1: N/A

4. Have the authors made all data underlying the findings in their manuscript fully available?

Reviewer #1: Yes

5. Is the manuscript presented in an intelligible fashion and written in standard English?

Reviewer #1: Yes

6. Review Comments to the Author

Reviewer #1: This extensively revised paper is now much improved and offers interesting insights into the value of brief psychotherapy. While clinical trials have shown the value of the latter, this method simply shows a probability of benefit compared with an alternative ‘placebo’ intervention. This means that many patients do not benefit from any intervention and it is therefore increasingly common for a qualitative study to be attached to clinical trials to better understand how and why some patients benefit and others don’t. This paper, then, fulfils that important role and I would support its publication without further amendments.

7. PLOS authors have the option to publish the peer review history of their article (what does this mean?). If published, this will include your full peer review and any attached files.

Reviewer #1: No

---

## [Editor Report · Acceptance letter]

5 Nov 2024

PONE-D-24-03765R1 

PLOS ONE

Dear Dr. Markussen, 

I'm pleased to inform you that your manuscript has been deemed suitable for publication in PLOS ONE. Congratulations! Your manuscript is now being handed over to our production team.

Kind regards, 

on behalf of

Dr. Ramune Jacobsen 

Academic Editor

PLOS ONE